# Microwave Irradiation vs. Structural, Physicochemical, and Biological Features of Porous Environmentally Active Silver–Silica Nanocomposites

**DOI:** 10.3390/ijms24076632

**Published:** 2023-04-01

**Authors:** Aleksandra Strach, Mateusz Dulski, Daniel Wasilkowski, Oliwia Metryka, Anna Nowak, Krzysztof Matus, Karolina Dudek, Patrycja Rawicka, Jerzy Kubacki, Natalia Waloszczyk, Agnieszka Mrozik, Sylwia Golba

**Affiliations:** 1Doctoral School, University of Silesia, Bankowa 14, 40-032 Katowice, Poland; aleksandra.strach@us.edu.pl (A.S.); oliwia.metryka@us.edu.pl (O.M.); 2Institute of Materials Engineering, Silesian Center for Education and Interdisciplinary Research, 75 Pulku Piechoty 1A, 41-500 Chorzow, Poland; sylwia.golba@us.edu.pl; 3Institute of Biology, Biotechnology, and Environmental Protection, Faculty of Natural Sciences, University of Silesia, Jagiellońska 28, 40-032 Katowice, Poland; daniel.wasilkowski@us.edu.pl (D.W.); anna.m.nowak@us.edu.pl (A.N.); agnieszka.mrozik@us.edu.pl (A.M.); 4Materials Research Laboratory, Silesian University of Technology, Konarskiego 18A, 44-100 Gliwice, Poland; krzysztof.matus@polsl.pl; 5Łukasiewicz Research Network, Institute of Ceramics and Building Materials, Cementowa 8, 31-938 Cracow, Poland; karolina.dudek@icimb.lukasiewicz.gov.pl; 6A. Chełkowski Institute of Physics, University of Silesia, 75 Pulku Piechoty 1, 41-500 Chorzow, Poland; patrycja.rawicka@us.edu.pl (P.R.); jerzy.kubacki@us.edu.pl (J.K.); 7Faculty of Chemistry, Silesian University of Technology, 44-100 Gliwice, Poland; natalia.szulc@polsl.pl

**Keywords:** porous silica spheres, silver nanoparticles, porosity, surface area, microbial activity, enzyme activity

## Abstract

Heavy metals and other organic pollutants burden the environment, and their removal or neutralization is still inadequate. The great potential for development in this area includes porous, spherical silica nanostructures with a well-developed active surface and open porosity. In this context, we modified the surface of silica spheres using a microwave field (variable power and exposure time) to increase the metal uptake potential and build stable bioactive Ag_2_O/Ag_2_CO_3_ heterojunctions. The results showed that the power of the microwave field (P = 150 or 700 W) had a more negligible effect on carrier modification than time (t = 60 or 150 s). The surface-activated and silver-loaded silica carrier features like morphology, structure, and chemical composition correlate with microbial and antioxidant enzyme activity. We demonstrated that the increased sphericity of silver nanoparticles enormously increased toxicity against *E. coli*, *B. cereus*, and *S. epidermidis*. Furthermore, such structures negatively affected the antioxidant defense system of *E. coli*, *B. cereus*, and *S. epidermidis* through the induction of oxidative stress, leading to cell death. The most robust effects were found for nanocomposites in which the carrier was treated for an extended period in a microwave field.

## 1. Introduction

Silver, for centuries, has been considered a miracle medicine to cure wounds and protect food from spoilage. The development of silver in nanometric form conceived as nanoparticles (Ag NPs) with a size smaller than 100 nm has set a new direction in microbiologically-active agents [1,2]. The positive biological aspect describes two different mechanisms of action (direct and indirect).

The direct approach explains cell death by changing the permeability of the cell membrane due to the adhesion of Ag NPs to the wall [3]. This mechanism is related to bacterial cell wall thickness [4]. The alternative hypothesis showed that a decrease in silver nanoparticle size leads to apoptosis due to phagocytosis and interactions with the protein structures of bacterial cells [5]. Silver nanoparticles also tend to denature the ribosome, inhibiting protein synthesis and impairing the function of bacterial β-galactosidase [6]. Ag NPs may affect the expression of enzymes involved in cellular oxidation due to the generation of reactive oxygen species (ROS) [7], leading to oxidative stress and DNA damage [6,8].

In turn, the indirect mechanism of action refers to the gradual release of Ag^+^ ions, with the degree of ion release depending on the development of the nanoparticle surface. Released silver ions covalently attach to the sulfhydryl groups of bacterial proteins, disrupting membrane transport and inhibiting enzyme activity [9]. The induction of oxidative stress by both silver nanoparticles and ions contributes to the development of antibiotic resistance by increasing the tolerance to oxidative stress. After repeated exposure to Ag NPs, pathogenic bacteria can produce adhesion proteins that cause Ag NP aggregation [10]. For this purpose, there are many developmental directions for new silver-based structures with extended antimicrobial effects that will limit bacterial resistance formation.

Other positive features of silver refer to long-time stability at normal conditions (‘slow aging’) [11,12] but relatively low at higher temperatures [13,14]. Silver aging and ion-release potential correlate with the particle size, oxidation rate [15,16,17,18,19] as well as synthesis conditions [10], whereby the oxidation is triggered by the contact of silver with an aqueous solution or higher temperature following in the sequence Ag(0) > Ag(I) > Ag(II) [20,21,22]. Silver oxidation at suitable water solutions and environmental conditions, including accessibility to atmospheric water vapor, promotes the crystallization of Ag_2_CO_3_ [23,24,25]. 

Silver carbonate crystallized in such conditions is considered a semiconductor with antibacterial and photocatalytic features. Similar to metallic or oxidized silver, silver carbonate exhibits features conducive to, e.g., the inactivation of *E. coli* both in limited accessibility of light and with a significant rise of the antimicrobial effect under visible light [26,27]. Similar effects were reported for *S. aureus* and *P. aeruginosa* under ambient light. Moreover, the antimicrobial effect resulted from releasing silver ions due to the dissolution of silver particles in the suspension or after visible light illumination [28,29]. Other applications of silver carbonate are focused on pollution removal, fuel production [30,31,32], and the photodegradation of rhodamine B (RhB) [33,34], methylene blue [35,36], and ethylparaben [37].

Unfortunately, despite biological-friendly effects, nanostructured silver features negative aspects distinctive for human health, animals, and ‘good’ microorganisms. The negative mechanism of silver resulted from the uncontrolled release of ions from nanoparticle surfaces and their strong tendency to aggregate. It provides a local increase in ecotoxicity or cytotoxicity, even at low metal concentrations [38]. Silver carbonate exhibits relatively poor stability and is prone to self-photo corrosion [39]. 

Therefore, stabilization of the silver or silver-based nanostructure is usually realized through core-shelled Ag_2_O/Ag_2_CO_3_ [39], Ag_2_CO_3_@Al_2_O_3_ [40], Ag_2_S@Ag_2_CO_3_ [41], or Ag_2_CO_3_@Ti_2_O [42] composites. Another promising solution is the immobilization of silver within the silicon dioxide (=silica). Silica, in this context, ought to be considered two-folded; the first positive aspect is related to high thermal and chemical stability, large surface area, and low density, while the second corresponds to its porosity [43,44,45,46]. Porosity is divided into micro- (<2 nm), meso- (2–50 nm), and macroporosity (>50 nm) [47,48]. Among them, microporous or mesoporous features poor mass transfer capacity and high surface areas that increase the metal-capturing potential. On the contrary, macroporous silica exhibits a high external mass transfer rate due to its interconnected larger pores at a relatively low specific surface area limiting metal capturing potential [49,50]. An increase in the macroporous silica’s functionality can be reached by chemical or microwave-assessed modification. Such surface treatment should prevent metal aggregation, improve suitable nanostructures’ stability and dispersion, and allow silver ions to be control-released [51,52]. As a result of surface modification, silver or silver-based nanostructures ought to be stabilized as Ag@SiO_2_, Ag_2_O@SiO_2,_ Ag_2_CO_3_@SiO_2,_ or even Ag_2_O/Ag_2_CO_3_@SiO_2_.

The paper summarizes a new insight into engineering environmentally applicable silica-based composites. For this purpose, a changeable microwave field and synthesis conditions follow the surface morphology—mainly the surface area, pore volume, and pore size. A structural modification was directed to increase the uptake potential of silica and stimulate stable, bioactive Ag_2_O/Ag_2_CO_3_ heterojunctions. The correctness of the synthesis assumptions was verified by analyzing the silver particle shape, size, and phase composition, as well as the molecular interaction of silver with silica. Morphological, structural, and chemical parameters were correlated with potential antimicrobial effects against reference Gram-positive and Gram-negative bacterial strains. Finally, the activities of three antioxidant enzymes, catalase (CAT), peroxidase (PER), and superoxide dismutase (SOD), were analyzed.

## 2. Synthesis

The powder silver–silica nanocomposite was fabricated during a chemical reaction in a colloidal suspension. For synthesis, silver nitrate (purity above 99.9999% with trace metal basis; Merck Life Science Sp.z.o.o., Darmstadt, Germany), sodium hydroxide (purity above 99.99%; Avantor Company, Gliwice, Poland), and commercially available spherical amorphous silica powder (S1: d = 15–20 nm, purity: 99.5+%, surface area: 170–200 m^2^/g, and true density: 2.4 g/cm^3^ [53]; and S2: d = 20–30 nm, purity: 99.5+%, surface area: 180–600 m^2^/g, and true density: 2.4 g/cm^3^ [54]), were used. 

In the first step, three individual flasks contained: 10 g of silica powder (S1 and S2) mixed with 200 mL of distilled water in addition to 2 g of silver nitrate and 2 g of sodium hydroxide dissolved in 20 mL of distilled water, each (Figure 1). Next, the colloidal silica suspension was subjected to microwave irradiation for 60 and 150 s in 150 and 700 W to activate the silica surface. Then, the water solution of sodium hydroxide was added dropwise to the flasks with the microwave-treated silica suspension and continuously stirred on a magnetic stirrer. Afterward, the water solution of the silver nitrate was added by Pasteur pipette drop by drop to the suitable flasks to fabricate the silver–silica colloids. The final products were individually filtered on a polyethylene filter, washed, and dried at room temperature to remove impurities. Finally, eight samples were prepared according to variable synthesis conditions (Table 1).

## 3. Results and Discussion

### 3.1. Morphology, Surface Parameters, and Particle Analysis

The nanocomposite particle diameters varied from d = 140 - 161 nm (Table 2, Figure 2). A statistically significant difference of d_H_ occurred depending on the type of silica (S1 or S2), and the microwave filed exposure parameters (P = 150 or 700 W and t = 60 or 150 s). Thus, Ag-S1 composites (AR1.1, AR1.3) featured higher hydrodynamic particle diameters than those corresponding to Ag-S2 (*** *p* < 0.001). 

Ag-S1 prepared at a microwave field of 700 W showed statistically higher values in the mean hydrodynamic diameter (d_H_) than Ag-S1 samples exposed to a microwave field of 150W (* *p* < 0.05). Conversely, Ag-S2 nanocomposites displayed the opposite trend of changes considering the microwave power (*** *p* < 0.001). Samples exposed to microwave fields of different power and at t = 150 s had no statistically significant differences. Higher values for silver-silica nanocomposites than those declared by the producer for reference silicas confirmed the silica surface activation due to microwave irradiation and the synthesis conditions. 

Moreover, strongly modified silica particle surfaces through over-melt of the edges led to the formation of aspherical (=rode-like) porously-altered structures. On the other hand, the effect of molten silica edges was enhanced by the catalytic properties of silver that stimulated surface modification. Nevertheless, the PDI lower than 0.4 yielded a monodisperse character of the samples with a low degree of aggregation in suspension (the Zeta potential values changed from −37 to −40 mV).

The synthesis condition yielded similar SSA_BET_ values independent of the silicas’ type (S1 or S2) (Table 3). Twice lower values for Ag-S1 and six times lower than for Ag-S2 confirmed a huge role of surface modification and the size of porously-altered structure on the sorption potential of S1 and S2 silicas. In this context, S2 silica seemed more sensitive to microwave irradiation and synthesis conditions than S1 (Table 3). Total pore volume (TPV) and porosity outcomes grouped the samples into two categories with a TPV value lower or higher than 1.0. The first group classified samples prepared at the shorter exposure of silica on microwave irradiation (AR1.1, AR1.3, AR1.5, and AR1.7), while the second group classified samples at a longer time (AR1.2, AR1.4, AR1.6, and AR1.8). The pore size diameter, D_pore_, changed between 30 and 50 Å. Lower pore size diameters resulted from the differentiated morphology of the silica, including higher internal pore wall roughness. However, the pore size values turned out to be much lower for the fabrication route at a longer time but independent from microwave irradiation power (Table 3).

Finally, hierarchical cluster analysis (HCA) considering SSA_BET_ and D_pore_ grouped samples into two or five clusters dependent on the Sneath criterion (Appendix A). Application of less restrictive criterion (66%) grouped samples among exposition time of silica on microwave irradiation (t = 60 or 150 s) into two clusters (AR1.1, AR1.3, AR1.5, AR1.7) and (AR1.2, AR1.4, AR1.6, AR1.8). According to a more restrictive criterion (33%), samples were divided additionally among irradiation time and power. Hence, three clusters separated nanocomposites synthesized at a shorter time and lower power: AR1.1 (red), AR1.5 (green), and higher power: AR1.3, AR1.7 (blue) from those of more extended time: AR1.2, AR1.6, AR1.8 (violet) and AR1.4 (yellow) (Appendix A). Samples belonging to violet and yellow clusters featured similar results illustrating a lower impact of power in this route. At the same time, D_pore_ values were much lower than for the fabrication route at a longer time but independent from microwave irradiation power (Table 3, Appendix A). A similar trend of changes was found for micropore volume and area, with lower values for samples fabricated at shorter times (Table 3). 

HCA analysis confirmed two independent routes of synthesis: (i) one with the silica surface activation at the default power and a longer time (t = 150 s) and (ii) the second obtained at a shorter time (t = 60 s) (Appendix A). The first route ensured a higher surface area with a smaller pore size, while the second one generally favored forming a system with a smaller active surface and a higher pore size (Table 3). In such a route, we additionally distinguished samples with the highest D_pore_ values (AR1.1 and AR1.5) and the lowest (AR1.4 and AR1.8), thus, suggesting that the most optimal parameters for surface modification were achieved at a short time (t = 60 s) and low microwave power (P = 150 W). Furthermore, synthesizing the Ag-Si1 and Ag-S2 nanocomposites was morphologically destructive, especially for S2. However, S1 and S2 silicas still demonstrated potential sorption towards silver capture. 

Proof of silver capturing success was provided through microscopic observations performed by SEM and TEM. SEM observations followed local chemical alterations on a macro scale, while TEM was used to observe morphological and chemical changes on a nanoscale level. According to the SEM-EDS results, the silver content was generally lower than 4.0 at.%, while the highest silver content was found for nanocomposites prepared at t = 150 s and P = 700 W. At the same time, nanostructures based on the S1 silica featured slightly higher silver content (Table 4). 

More local TEM-EDS observations revealed higher silver content for samples prepared at t = 60 s and P = 150 W with a low silver concentration for silicas microwave-irradiated for a longer time (Table 4). Furthermore, S1 and S2 silicas, regardless of microwave treatment (power and time), permitted the crystallization of heterogeneously distributed spherical or aspherical silver particles (Figure 3 and Appendix A). Silicas treated through high microwave irradiation (AR1.3, AR1.4, AR1.7, and AR1.8) featured the highest number of aspherical particles. In comparison, no aspherical Ag NPs were found for a low microwave field and short time (Table 5). Ag-S1 and Ag-S2 highlighted similar shape-forming tendencies, except for Ag-S2 prepared at P = 700 W and t = 60 s with a higher percentage of aspherical particles (Table 5). A difference in particle shape resulted in the necessity of data limitation and calculation of the average particle size considering only the fully spherical shape. As a result, the particle size distribution histogram was fitted using a lognormal function (logN) with values changing from d = 1.72 to 6.31 nm (Figure 3, Table 5, and Appendix A). 

The diffraction pattern revealed cell parameters a_0_ = 4.085(7) Å and space group Fm-3m. Inter-planar d-spacing indicated higher phase differentiation, i.e., the face-cantered cubic silver, the monoclinic AgO, and trigonal Ag_2_O (Figure 3). TEM data concluded that longer microwave field exposure time led to the crystallization of Ag NPs with a larger diameter. Furthermore, a lower microwave field (P = 150 W) formed smaller and spherical Ag NPs. In contrast, the P = 700 W enforced the crystallization of larger and more aspherical silver nanoparticles.

### 3.2. Structural Analysis

The X-ray diffraction measurements revealed the co-association of amorphous with a broad hump at 2θ up to 30°, and crystalline phases revealed two polymorphic forms of silver carbonate (Ag_2_CO_3_) (Figure 4). One crystallized in a hexagonal crystal system (P3_1_c), while the second had a monoclinic structure (P2_1_/m).

The lattice parameters determined from the Rietveld refinement are summarized in Table 6 for individual phases. Under different synthesis conditions, minor shifts of the diffraction lines and differences in the lattice parameters indicate a disturbed structure of silver carbonates due to phase crystallization. Structure disturbances in the crystal lattice resulted from the incorporation of foreign ions or the presence of vacancies. In all samples, the amorphous phase was the predominant one.

However, considering the diffraction peaks’ intensity, the nanocomposites differ in the content of polymorphic silver carbonates. In addition, a more significant proportion of crystalline phases was observed in the samples produced in a short duration of microwave irradiation. The diffraction peaks of silver and silver oxides were not identified due to their low content or nanocrystalline nature.

The XRD results provided incomplete information about the silica structure and mutual interactions between individual nanocomposite materials. As a result, the XRD outcomes were supplemented by Raman spectroscopy. Regardless of the type of silica, the microwave treatment and synthesis conditions provided extra information on the silica network (Figure 5). The introduction of silver by synthesis combined with the morphological change and the formation of porous rod-like structures forced the formation of a disordered silica network with numerous NBOHC, E’, or neutral oxygen center structural defects marked by high-intensity features at 278 and 323 cm^−1^ [55,56,57]. 

The series of variable point defects is comparable to Ag-S1 and Ag-S2 composites and independent of the microwave field, including the power and time. Deformation of the silica lattice also revealed the appearance of intermediate-order silica superstructures in the form of bands around 382, 458, and 693 cm^−1^ originating from O-Si-O bending vibrations within n-member rings with n > 4 [58,59] and movement within the Si_2_O_7_^−^ unit [60] (Figure 5).

The determination of the degree of depolymerization, structural ordering, and silica potential to capture metal ions determine the number and type of *Q^n^* units (n = 0–4 and denotes the number of bridging oxygen (BO) per tetrahedral SiO_4_) (850–1250 cm^−1^) [61,62,63,64,65,66,67]. As a result, the molecular image of Ag-S1 and Ag-S2 nanocomposites, regardless of the synthesis conditions, consists of three Gaussian peaks of similar intensity around 895, 945, and 1005 cm^−1^ associated with Si-O* stretching modes in *Q*^1^, *Q*^2^, and *Q*^3^ units with three, two, and one non-bridging oxygen (NBO), respectively. 

Moreover, the Raman spectra of AR1.2, AR1.4, AR1.6, and AR1.8 are characterized by additional bands around 540, 1025, and 1100 cm^−1^, originating from Si-O* vibrations within the metal-activated *Q*^3^ unit (Figure 5) [63,68,69]. According to the physicochemical and morphological studies, silicas with larger pores featured a capillary effect with tremendous potential to capture more significant amounts of metal, which led to the formation of stronger binding between silver nanoparticles and the interior of silica pore walls (Table 2 and chemical analysis).

A signal of silver oxide appeared after the fitting procedure as three well-separated bands at 233, 417, and 490 cm^−1^ (Figure 5). In contrast, the band arrangement corresponds to the oxide with the crystallographic structure closely related to Ag^I^Ag^III^O_2_ [70,71]. Consequently, the band at 488 cm^−1^ comes from Ag(I) cations linearly coordinated by oxygen in the AgO structure, while at 417 cm^−1^, the vibration is within the square-planar Ag^III^O_4_ unit. The lowest-laying band at 233 cm^−1^ is derived from the symmetric bending mode of the Ag^III^O_4_ unit [72,73], while Raman’s active asymmetric stretching mode overlaps with the O-Si-O motion. 

Some differences in the band positions resulted from the presence of non-bridging silica bonds that enforced the formation of Si-O-Ag stabilizing the nanoparticles within the carrier. The bands at 709 and 1068 cm^−1^ arose from the silver carbonate [74,75,76]. Two separated CO_3_^2-^ stretching vibration-related bands occurred between 1350 and 1650 cm^−1^, confirming the silver carbonate with an aragonite-type structure [77]. One idea to explain two different forms of silver was the assumption of a core-shell structure formation [78]. 

Another idea illustrated a crystallization of silver oxide (Ag_2_O) and silver carbonate (Ag_2_CO_3_) simultaneously. In this approach, Ag_2_O had to be pulled through capillary action into the pore and stabilized inside the silica carrier through unsaturated bonds. At the same time, silver carbonate crystallized immediately only at surface-bonded silver oxide with great accessibility of atmospheric CO_2_. Both hypotheses proved the molecular sorption potential of S1 and S2 (Figure 5).

### 3.3. Surface Chemical Studies

X-ray photoemission spectroscopy (XPS) was performed to determine the surface atomic concentration (Table 7) and chemical states of silver, oxygen, silicon, and carbon (Figure 6 and Figure 7).

Generally, the deconvoluted Ag 3d core lines of Ag-S1 (Figure 6) and Ag-S2 (Figure 7) consist of two or three doublets with binding energies around 368.3, 368.1, 368.2, and 367.6 eV. The maxima of individual peaks point to Ag, AgO, Ag=O, and Ag_2_CO_3_, wherein unambiguous separation of metallic and silver oxides is difficult because of the close binding energy values [79]. A clue confirming metallic nanoparticles is the signal originating from non-stochiometric silver (Ag_x_O_y_) with binding energy from 366.1 to 366.9 eV due to the adsorption of different silver oxygen structures [80]. 

The relative atomic concentration of silver differs from 0.10 up to 0.69 at.%. Nanocomposites exposed longer to microwave irradiation had lower silver capture potential (Table 7). Conversely, the highest uptake potential yielded silver–silica nanocomposites fabricated at P = 700 W and t = 60 s with 0.69% at.% for Ag-S1. From a chemical point of view, the optimal development of the silica surface is stated to be with high power and a short microwave irradiation time.

The chemical states observed during the analysis of the Ag 3d core line correspond to the chemical state observed for the O 1s line of AgO (530.0 eV), Ag_2_O (528.8 eV), SiO_2_ (532.7–532.9 eV), and non-stoichiometric SiO_x_ (531.9 eV) (Figure 6 and Figure 7). Interestingly, in practice, the XPS results as a surface-sensitive technique do not provide chemical evidence of silver carbonate, regardless of the more bulk-sensitive XRD. Hence, these results indirectly suggest forming a core-shell system with a silver oxide layered by silver carbonate. The XPS and XRD data proofed assumed heterojunction silver structured composites. Microwave irradiation application caused the absorption of ambient carbon and the sorption or desorption of oxygen by silicon. As a result, the Si 2p core level fitting revealed the line corresponding to SiC, SiO_x,_ or SiO_2-x_ (Figure 6 and Figure 7). The surface carbon formed C-C/C-H bonds, Si-C, CO^+^O_2/_Ag (284.8eV), O-C-O (288.3 eV), C-OH (285.8–286.7 eV), C=O (287.3–287.6 eV), and O-C=O (288.9–289.3 eV) (Figure 6 and Figure 7).

## 4. Biological Studies

### 4.1. Microbial Activity of Newly Synthesized NCs

The antibacterial activity of all the Ag-SiO_2_ nanocomposites was evaluated against three reference bacterial strains: Gram-negative *Escherichia coli* (ATCC^®^ 25922™), and Gram-positive *Bacillus cereus* (ATCC^®^ 11778™), *Staphylococcus epidermidis* (ATCC^®^ 12228™). The model bacteria proved sensitive to all tested NCs—in particular, the silver nanoparticle responses were found to be strain-specific silver nanoparticle responses (Table 8). Generally, *B. cereus* was characterized by the highest sensitivity to NCs, for which the average MBC values ranged from 5 to 10 mg L^−1^. 

Based on the calculated IC_50_ values, the toxic effect for *B. cereus* decreased in the following order: AR1.6 > AR1.3 > AR1.1 > AR1.7 > AR1.2 > AR1.5 > AR1.4 > AR1.8. The opposite effect was observed for *E. coli* and *S. epidermidis*, which proved to be more resistant to stress caused by NCs. Interestingly, among all studied bacterial strains, AR1.8 and AR1.6 had the highest lethal effect on the *S. epidermidis* cells with IC_50_ = 0.41 and 1.69 mg L^−1^, respectively. 

Comparing the IC_50_ results generated for Gram-negative *E. coli*, the decreasing bactericidal influence of the tested NCs can be ordered as follows: AR1.6 > AR1.8 > AR1.3 > AR1.2 > AR1.4 > AR1.1 > AR1.7 > AR1.5. A unique and different response to the synthesized NCs characterized all bacterial strains. Nevertheless, all studied NCs were highly toxic to *E. coli*, *B. cereus*, and *S. epidermidis* (MBC 2.5–25 mg L^−1^), with AR1.6 as the most toxic nanocomposite to all three strains. Its high toxicity yielded small-size silver nanoparticles that were likely low-bound to the silica carrier, providing free passage through the outer layers of microbial cells. 

Furthermore, the percentage reduction in the spherical shape of Ag NPs can increase the porosity of tested Ag-SiO_2_ and further enhance the contact of NCs with the surface of bacterial strains compared to completely spherical Ag NPs. This property of synthesized nanocomposites may also increase their biological activity due to the higher surface-to-volume ratio [81,82,83]. 

Previous studies have indicated that inorganic NPs with uneven surfaces and irregular shapes are characterized by more significant activity, which could be attributed to the faster release of metal ions from the surface [81,84,85]. This phenomenon is demonstrated by the collected results, as spherical AR1.1 and AR1.2 proved to be less toxic than the rest of Ag-SiO_2_. Considering the d_H_, the homoagglomeration of Ag-SiO_2_ nanocomposites should be comparable for all silver–silica nanocomposites. 

However, the presence of compounds in the culture medium, bacterial secretions, and the bacteria themselves may alter the toxicity of the tested nanocomposites [86,87]. Hence, weaker electrostatic interaction of the agglomerates with the surface of bacterial cells explains the lower lethal effect of AR1.2 compared to other nanostructures. 

Similarly, a more toxic effect of Ag NPs was found for triangular nanostructures on *Escherichia coli* and *Pseudomonas aeruginosa* cells regarding spherical nanoparticles [88]. Other studies reported the opposite effect, i.e., spherical Ag NPs had a more significant bactericidal effect on both Gram-negative (*Klebsiella pneumoniae* and *Pseudomonas aeruginosa*) and Gram-positive (*Bacillus subtilis* and *Staphylococcus aureus*) bacteria compared with NCs with a rod shape [89]. 

Acharya et al. [90] demonstrated that Ag@SiO_2_ core-shell nanoparticles (~17 nm) had a more substantial antibacterial effect on *B. subtilis*, *S. aureus*, *Serratia marcescens,* and *K. pneumoniae* than singular Ag NPs (~14 nm) due to controlled and sustainable release of Ag(I). Correspondingly, Gankhuyag et al. [91] and Pal et al. [92] recorded an enhanced bactericidal activity of SiO_2_@AgNPs and Ag-NP hybrid silica films against *E. coli* cells, respectively. A similar effect was reported for the core-shell silver–silica nanosystem against highly resistant *Mycobacterium tuberculosis* [93]. The so-called "Trojan horse effect" as a biocidal mechanism of AR-nanocomposites can be assigned to the results of the presented work. 

Furthermore, releasing active Ag^+^ ions from the SiO_2_ carrier as an effective antibacterial agent would explain their highly toxic effects on tested microorganisms [81]. It also suggests that depositing or capping Ag NPs in core-shell systems increases their toxicity, enabling their stabilization and preventing the direct formation of Ag-NP aggregates [94]. These findings also follow Ahmed et al. [95], who observed a significant toxicological impact of Ag NPs obtained through microwave-mediated synthesis against *E. coli*, *P. aeruginosa*, *S. aureus,* and *K. pneumoniae*. 

### 4.2. Antioxidant Enzymes Activity in Response to NC Stress

The activity of the triad antioxidant enzymes was measured to examine the effect of the tested NCs on a catalytic antioxidant cell system, including catalase (CAT) and peroxidase (PER), where breakdown generated O_2_ and H_2_O from H_2_O_2_. Furthermore, the third examined enzyme was superoxide dismutase (SOD), catalyzing the dismutation of O_2_**^·^**^−^ to O_2_ and H_2_O_2_. The obtained findings showed unique catalytic profiles for each bacterial strain and the explicit dependency of the primary antioxidant defense system on the properties of Ag-SiO_2_ (Figure 8). 

Generally, the treatment of *E. coli* with AR1.1, AR1.3, AR1.5, and AR1.7 had a stimulating effect on CAT activity, especially in the presence of AR1.1 (23.61 U mg^−1^). Successively, all nanocomposites had a significant enhancing impact on PER activity except AR1.2. For instance, the addition of AR1.7 caused the highest increase (2.3 fold) in PER activity (6.04 U mg^−1^) compared to the control. Intriguingly, AR1.3 and AR1.7 were the only structures that stimulated SOD activity in *E. coli* cells, while inhibition of its functioning was observed in the presence of remaining Ag-S1 and Ag-S2.

In the case of the *B. cereus* strain, the NC treatment manifested diverse effects on CAT, PER, and SOD activities (Figure 9). Among all tested NCs, only AR1.1 and AR1.5 stimulated CAT and SOD activity, resulting in 13- and 21.5-fold as well as 3- and 5.5-fold increases compared to the control, respectively. In contrast, the conducted research revealed an inhibitory effect of AR1.2, AR1.4, AR1.6, and AR1.8 on CAT and SOD activities, while these nanostructures had the opposite impact on PER functioning. 

Treatment of *S. epidermidis* with individual NCs demonstrated their divergent effects on the activity of CAT, PER, and SOD (Figure 10). We found that the presence of AR1.5 and AR1.8 caused an increase in CAT activity. In contrast, AR1.1, AR1.3, and AR1.4 treatments resulted in a decrease in CAT. Compared to the control, the most significant inhibitory effect (10-fold lower) was observed after exposure to AR1.4. 

In turn, treatment of *S. epidermidis* with AR1.1 and AR1.7 resulted in the highest (3-fold) increase in PER activity, while the addition of AR1.4 caused the most significant decrease (1.6-fold). Interestingly, the application of Ag-SiO_2_ increased the SOD activity, particularly AR1.4, which had the most distinguishable effect on its functioning.

The collected data for the catalytic profiles of the three strains indicate that the Ag-SiO_2_ nanocomposites altered the antioxidant defense systems of *E. coli*, *B. cereus,* and *S. epidermidis* compared to the control. The observed changes in the CAT, PER, and SOD activity can be ascribed to the induction of oxidative stress in bacterial cells. Previous studies have shown that inorganic Ag NPs can generate reactive oxygen species (ROS), such as hydroxyl radicals (^•^OH), hydrogen peroxide (H_2_O_2_), or singlet oxygen (^1^O_2_) [81,96]. Despite conducting intensive research into the effect of nanoparticles on the antioxidant system of microorganisms, the current research concerns only selected strains, enzymes, and specific nanostructures. 

Therefore, it is crucial to explore this subject further to understand the characteristics of the biological activity of engineered nanostructures. Analysis of the collected results shows that Ag-SiO_2_ could generate and increase the level of H_2_O_2_ in bacterial cells due to the noticeable and significant changes in CAT and PER activities, resulting in stimulation of the antioxidant defense system. Both Ag NPs and the released metal ions can generate ROS but can also interact with protein groups modifying their functioning, consequently causing their inactivation [97,98,99]. 

The biochemical data would explain the significant decrease of both CAT and PER, especially in *B. cereus* and *S. epidermidis* cells, after treatment with selected Ag-SiO_2_. However, SOD activity in all three strains was the least affected by the nanocomposites, indicating the active defense systems of *E. coli*, *B. cereus,* and *S. epidermidis* against ^1^O_2_ generation. AR1.3 and AR1.7 in *E. coli* culture resulted in a characteristic pattern of stimulated antioxidant activity of CAT, PER, and SOD. 

This phenomenon could be explained by the stabilized and increased induction of ROS previously mediated by irradiation of AR1.3 and AR1.7 with more substantial microwave power. It has been reported that radiosensitization and other chemical phenomena applied in the synthesis of nanomaterials may also affect ROS synthesis in biological systems [100]. Nevertheless, this pattern is not reproducible in other strains, which confirms the dependence of the observed changes on the physicochemical properties of nanocomposites and the unique characteristics of microorganisms. Furthermore, the increased toxic effects of AR1.2, AR1.4, AR1.6, and AR1.8 on SOD and CAT in *B. cereus* cultures may be due to the protein denaturation and elevated concentration of H_2_O_2_.

Similarly, Yuan et al. [101] observed a considerable inhibition of SOD and CAT in *P. aeruginosa* and *S. aureus* exposed to Ag NPs, which positively correlated with ROS generation and plausible inactivation of enzyme activities. By comparison, Liao et al. [99] recorded decreased CAT, PER, and SOD activity in *P. aeruginosa* cells in the presence of Ag NPs. In different experiments, Jain et al. [102] demonstrated a positive correlation of an elevated ROS generation with increased SOD and PER activities in *E. coli*, *P. putida*, *B. cereus,* and *S. aureus* exposed to Ag NPs. Similarly, a considerable increase in the CAT activity of *E. coli* and *B. cereus* was recorded in the presence of SiC/Ag/CE/T nanocomposites [103]. Our collected data showed antioxidant activities are a strain-specific trait and depend on the type of nanomaterial.

### 4.3. Statistical Data Exploration

Cluster analysis and PCA were performed for the physicochemical and biological parameters to evaluate NC treatment variability and determine the relationships with the whole dataset. The results from PCA analysis explained 82.11%, 79.05%, and 69.66% of the data variability for *E. coli*, *B. cereus,* and *S. epidermidis*, respectively (Figure 11). Two clusters were distinguished for each strain for the correlation biplot projection along PC1.

Similar compositions of the projections were observed for the tested strains, where AR1.1, AR1.3, AR1.5, and AR1.7 were arranged into one plot, whereas the second cluster contained AR1.2, AR1.4, AR1.6, and AR1.8 (Figure 11A,C,E). The coordination biplot for *E. coli* revealed a strong negative correlation of PER with SSA_BET_ and CAT with V_P_ and S_P_ (Figure 11B). In turn, for *B. cereus*, the most distinguishable negative correlation was observed for the silver particle diameter (D_Ag_), CAT, and SOD (Figure 11D). The obtained PCA projection for *S. epidermidis* demonstrated a strong negative correlation of SOD with the number of particles with a spherical shape (NP_sf_). At the same time, the other parameters were more scattered (Figure 11F).

The cluster analysis displays a characteristic relationship between the obtained results and NC treatments, providing thematically similar patterns for all tested strains (Figure 12). The dendrogram projections obtained for *E. coli*, *B. cereus,* and *S. epidermidis* revealed that SSA_BET_, D_pore,_ and S_P_ were the most discriminating analyses. At the same time, among the measured enzymes, the most differentiating one was CAT. According to the acquired dendrogram, the most characteristic and discriminating were AR1.5 for the *E. coli* and *B. cereus* strains, while AR1.4 had the most divergent effects on *S. epidermidis*.

## 5. Materials and Methods

### 5.1. Structural Analysis

An X’PertPro MPD PANalytical X-ray diffractometer (Malvern PANalytical, Almelo, the Netherlands) with a Cu *K_α_* radiation (λ = 1.54 Å) was used for analyzing the crystal structure of Ag-SiO_2_. The Rietveld method refined the lattice parameters using HighScore Plus (version 5.1, Malvern PANalytical, Almelo, The Netherlands) software and the ICCD PDF-4+ 2023 database.

The three-dimensional silica network was followed through a WITec confocal Raman microscope CRM alpha 300 R (WITec Wissenschaftliche Instrumente und Technologie GmbH, Ulm, Germany) equipped with an air-cooled solid-state laser (λ = 532 nm). The excitation laser radiation was coupled into a microscope through a polarization-maintaining single-mode optical fiber with a 50 μm diameter. The monochromatic light was focused on the sample by an air Olympus MPLAN (100x/0.90NA) objective. Raman scattered light was passed through a multi-mode fiber (50 μm diameter) into a monochromator with a 600 line/mm grating and a CCD camera.

The spectrometer monochromator was checked before the measurements using a silicon plate (520.7 cm^−1^). Raman spectra were accumulated over 20 scans with an integration time of 20 s and a resolution of 3 cm^−1^. The post-processing analysis, such as baseline correction and cosmic ray removal, was performed in the WITecProjectFive Plus (version 5.1.1, WITec Wissenschaftliche Instrumente und Technologie GmbH, Ulm, Germany). The peak fitting analysis was performed in GRAMS (version 9.2, Thermo Fisher Scientific, Waltham, MA, USA) software.

### 5.2. Morphological Studies and Chemical Composition

TEM micrographs with chemical imagining were collected with a probe Cs-corrected S/TEM Titan 80-300 FEI microscope equipped with an EDAX EDS detector. The images were recorded in STEM mode, using the HAADF (high-angle annular dark field) and bright-field (BF) detectors. At the same time, the local chemical content and chemical mapping were realized using an Energy Dispersive Spectrometer (EDS). A scanning electron microscope equipped with an energy dispersive X-ray spectrometer (SEM-EDS) (Phenom ProX (ThermoFisher, Eindhoven, The Netherlands) at an accelerating voltage of 15 kV was used to estimate the chemical content in the microscale.

The X-ray photoelectron spectroscopy (XPS) measurements were made using a Physical Electronics PHI 5700 spectrometer (Physical Electronics Inc., Chanhassen, MN, USA). Photoelectrons were excited by monochromatized Al Kα radiation from the sample surface. The survey spectrum showed the presence of main core level lines from C, O, Si, and Ag with no evidence of impurities. The high-resolution XPS spectra for all core levels were corrected using the Iterated Shirley algorithm for the background signal. At the same time, the bands were fitted by a composition of Gaussian and Lorentzian lines in Multipack software. The binding energy value was corrected for minor surface charging effects by referring to the C 1s line at 284.5 eV.

### 5.3. Estimation of Nanoparticles Size

The diameter of the nanoparticles was measured at 25 °C on a Zetasizer Nano ZS (Malvern Panalytical, Grovewood Road, Malvern, UK). The analyzer was equipped with a He-Ne laser wavelength of 633 nm. The particle diameter was measured using dynamic light scattering (DLS), allowing us to obtain the diffusion coefficient (D) and the hydrodynamic diameter using the Stokes–Einstein equation (1). The measurements were performed in polystyrene cuvettes with an optical path of 1 cm at a detection angle of 173°. 

Before measurement, all samples were diluted in water and sonicated with an ultrasonic homogenizer (Omni Sonic Ruptor 400, PerkinElmer, Kennesaw, GA, USA). The sonication time was repeated thrice for 5 min (continuous mode; ultrasonic power did not exceed 20%) until a homogeneous suspension, and repeatable results were obtained. Each sample was measured ten times, and the results were averaged. The averaged values of hydrodynamic diameter (d_H_) and polydispersity coefficient (PDI) are summarized in Table 2.
(1)$$d_{H}=\frac{kT}{3\pi \eta D}$$
where *d_H_*: hydrodynamic diameter, *k*: Boltzmann’s constant, *T*: absolute temperature, *η*: viscosity of the diluent, and *D*: diffusion coefficient.

### 5.4. Determination of Nanoparticle Surface Charge

The Zeta potential was measured on Malvern’s Zetasizer Nano ZS (Malvern Panalytical, Grovewood Road, Malvern, UK) at 25 °C and in a U-shaped cuvette (DTS1070). The detection angle was 173°. The electrophoretic mobility (U_E_) was determined using laser Doppler velocimetry (LDV), wherein Zeta potential was recalculated using Henry’s equation (2). Samples were prepared analogously to the sample preparation procedure for measuring the diameter. Each sample was measured five times, and the average values and standard deviations are shown in Table 2.
(2)$$U_{E}=\frac{2\varepsilon zf\left(Ka\right)}{3\eta }$$
where *U_E_*: electrophoretic mobility, *ε*: dielectric constant, *z*: Zeta potential, *f(Ka)*: Henry’s function, and *η*: viscosity.

### 5.5. Determination of the Surface Area, Pore Volume, and Pore Diameter

The specific surface area (SSA_BET_), micropore volume (V_p_), area (S_P_), and pore size (D_pore_) were determined using a Gemini VII 2390a analyzer (Micromeritics Instruments Corp., Norcross, GA, USA). The measurements were performed at the boiling point of nitrogen (−196 °C), and final data was obtained using the Brunauer–Emmet–Teller (BET) method and *t*-plot analysis. The total pore volume (TPV) was calculated from the gas sorption isotherm at p/p_0_ close to saturation pressure (0.995 p/p_0_). 

The porosity was calculated as the TPV ratio to the sum of the TPV and solid particle volume (Table 3). Samples, before the measurements, were thermal-treated at 250 °C for three hours to remove gases and vapors adsorbed on the surface during the synthesis. This analysis was achieved with a VacPrep 061 degassing system (Micromeritics Instruments Corp., Norcross, GA, USA). Samples were not analyzed immediately after the degassing procedure was kept at 60 °C. The instrument’s accuracy was verified for each use by analyzing a carbon black reference material of known surface area (P/N 004-16833-00 from Micromeritics, Norcross, GA, USA).

### 5.6. Microbial Studies

#### 5.6.1. Bacterial Strains and Toxicological Studies

The assessment of microbial response to silver-silica nanocomposites (NCs) was performed using bacterial strains purchased from the American Type Culture Collection (ATCC) comprising Gram-negative *Escherichia coli* (ATCC^®^ 25922™) and Gram-positive *Bacillus cereus* (ATCC^®^ 11778™) and *Staphylococcus epidermidis* (ATCC^®^ 12228™). 

The minimum bactericidal concentration (MBC) and half-maximal inhibitory concentration (IC_50_) were estimated using the dilution method. For this purpose, standard solutions (200 mg L^−1^) with individual NCs were suspended in a sterile H_2_O_MP_ (ultrapure Millipore water) and sonicated before application using Vibra-Cell™ at 20 kHz for 10–20 min. Primarily, the lysogeny broth (LB) medium (tryptone 10 g L^−1^, NaCl 10 g L^−1^, and yeast extract 5 g L^−1^) was inoculated with a bacterial suspension in 0.85% NaCl-containing cells. The logarithmic growth phase was achieved until optical density OD_600_ = 0.1 (~10^7^ CFU mL^−1^). 

Subsequently, NCs were added to the bacterial cultures with final concentrations ranging from 0.1 to 100 mg L^−1^ and were incubated at 37 °C with shaking conditions (140 rpm). The control samples in this research were bacterial cells cultured without NCs. Following 24 h incubation, dilution series of each culture were prepared in 0.85% NaCl, subcultured on LB agar plates, and incubated for 24 h at 37 °C. Afterward, bacterial colonies were counted and expressed as colony-forming units (CFU mL^−1^). The lowest concentration of an antimicrobial agent, defined as MBC, was established as a 100% mortality rate of bacteria [104,105], whereas IC_50_ values of each NCs were estimated with the AAT Bioquest calculator [106].

#### 5.6.2. Antioxidant Enzymes Response to NC Treatment

The influence of NCs on the antioxidant cell defense system was examined by analyzing the activity of three antioxidant enzymes: catalase (CAT), peroxidase (PER), and superoxide dismutase (SOD) in *E. coli*, *B. cereus,* and *S. epidermidis* cells. For this purpose, bacteria were exposed to NCs at concentrations equal to IC_50_ values for 24 h (at 37 °C and 140 rpm). 

Enzymes were isolated from overnight cultures through Hegemans’ method [107]. Then, bacterial cultures were centrifuged (4 °C, 5000 rpm, and 20 min), and the obtained pellet was suspended in 50 mM phosphate buffer (pH 7.0). Afterward, the enzymes were released from cells during sonication (six times for t = 15 s with t = 30 s intervals and f = 20 kHz). The supernatant obtained after centrifugation (4 °C, 12 000 rpm, and 20 min) was subjected to the study of microbial antioxidants. 

The activity of CAT was assessed spectrophotometrically based on H_2_O_2_ decomposition—measured as a decrease in the absorbance at λ_240_ in time. The specific enzyme activity was calculated using the molar extinction coefficient (36,000 dm^3^ mol^−1^ cm^−1^). The Sigma-Aldrich colorimetric protocol based on the measurement of an absorbance intensification at λ_420_ equivalent to the increase of the purpurogallin product was applied to evaluate the PER activity. 

Successively, SOD activity was determined by the reduction of tetrazolium salt and a decrease in the dye’s color intensity (λ_450_) using a commercial reagent kit (cat. 19160; Sigma-Aldrich, St. Louis, MI, USA). The specific activity of SOD was calculated according to Zhang et al. [108]. The samples’ total protein concentrations in the isolated protein fractions were quantified using the Coomassie Brilliant Blue G-250 reagent method [109]. Finally, the CAT, PER, and SOD activities were expressed as U mg^−1^ of protein.

### 5.7. Statistical Analysis

Hydrodynamic particle size diameter was repeated in ten independent measurements for each sample. SEM-EDS and TEM-EDS measurements were done in three and five individual places. The toxicological and antioxidant enzyme response data were repeated three times. Data were statistically treated and presented as mean ± standard deviation values (± SD) each time. The statistical significance between the given Ag-S1 and Ag-S2 NCs and mean hydrodynamic diameter or their effects on enzyme activities were followed using a one-way ANOVA and Tukey’s Honest Significant Difference test (HSD). The substantial variations in experimental groups were represented in figures by annotated letters as *p* < 0.05 (*) and *p* < 0.001 (***) levels of significance. All experimental data were subjected to multivariate analysis. Hydrodynamic particle size was graphically presented using box-and-whiskers graphs. The specific surface area was correlated with pore size diameter by applying hierarchical cluster analysis (HCA) at the usage of the furthest neighbor, Euclidean distance, and application of the Sneath criterion (33% and 66%). In turn, the multifaceted analysis covering various aspects of the influence of NCs on the first line of the antioxidant defense system was performed using principal component analysis (PCA). The projection of the topographic map by comparing an entire set of physicochemical and biological variables was used to establish similar groups of objects forming clusters.

All the graphical/statistical processing data were estimated using MS Office 2019 (Microsoft Inc., Redmond, WA, USA), OriginPro2023 (OriginLab Corporation, Northampton, MA, USA), and the Statistica 13.3 software package (TIBCO Software Inc., Palo Alto, CA, USA).

## 6. Conclusions

The presented outcomes demonstrated the influence of different microwave irradiation conditions on modifying the absorption of silver-based nanostructures captured on silica carriers. Detailed studies showed that microwave treatment modified the silicas morphologically, increased the carrier’s shape and open porosity, and lowered the active surface. 

These effects became stronger and observable through a shorter exposure to microwave irradiation. However, modifying the surface of silica spheres using a microwave field both at t = 60 and 150 s, as well as P = 150 and 700 W, allowed for silver uptake and its crystallization as stable bioactive Ag_2_O/Ag_2_CO_3_ heterojunctions. Based on the statistical approach, the applied silica surface modification procedures divided the results into two routes in which the Ag-SiO_2_ systems were significantly different, physicochemically and structurally. The power of the microwave field proved to have a minor effect on carrier modification compared with the time (t = 60 or 150 s). We showed that increased time and power of microwave irradiation gradually increased the size of the silver particles. Increasing the microwave irradiation time and power enhanced the silver nanoparticle’s asphericity. Finally, an extended exposure time to the microwave field corresponded with lower porosity and content of silver captured by the silica. 

The surface-activated and silica carriers with Ag_2_O/Ag_2_CO_3_ were used to study antimicrobial activity and the effects on antioxidant enzyme functioning. We found that a greater sphericity of silver nanoparticles resulted in more considerable toxicological impacts against *E. coli*, *B. cereus*, and *S. epidermidis*. Furthermore, such structures affect the antioxidant defense system of *E. coli*, *B. cereus*, and *S. epidermidis* through enzymatic activity, causing the induction of oxidative stress and leading to cell death. The most robust effects were found for nanocomposites in which the silica carrier was treated for an extended period in a microwave field.

## Figures and Tables

**Figure 1 ijms-24-06632-f001:**
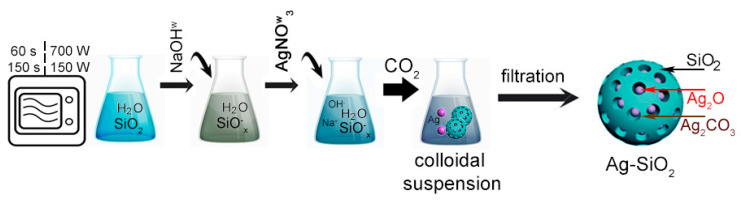
The synthesis of silver–silica spherical nanocomposites.

**Figure 2 ijms-24-06632-f002:**
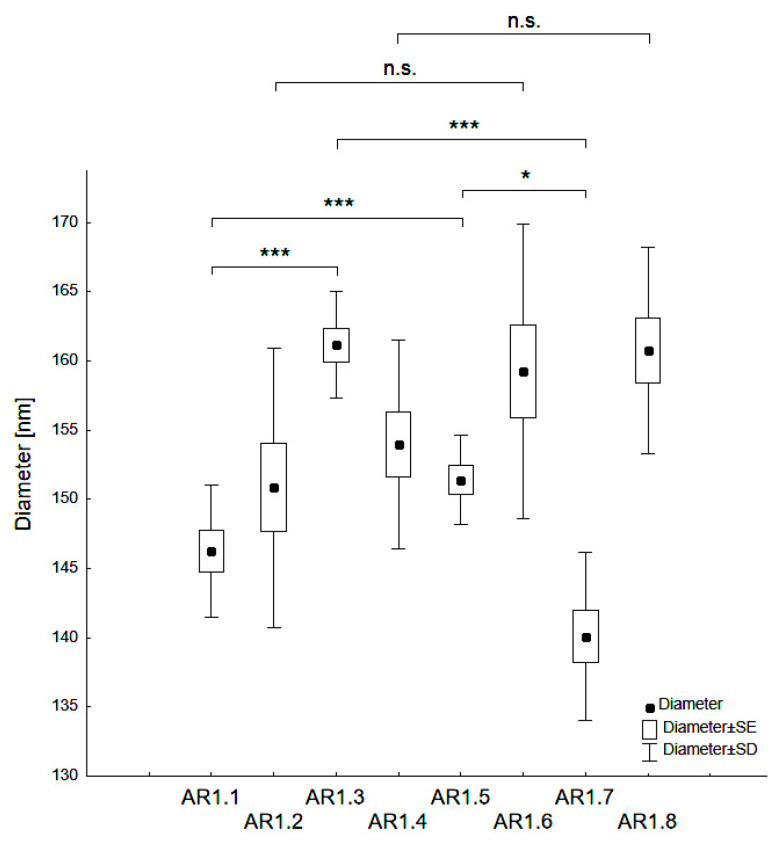
Categorized box-and-whisker plot for particles of Ag-SiO_2_ nanocomposites prepared at variable conditions with a marked level of significance (* *p* < 0.05 and *** *p* < 0.001).

**Figure 3 ijms-24-06632-f003:**
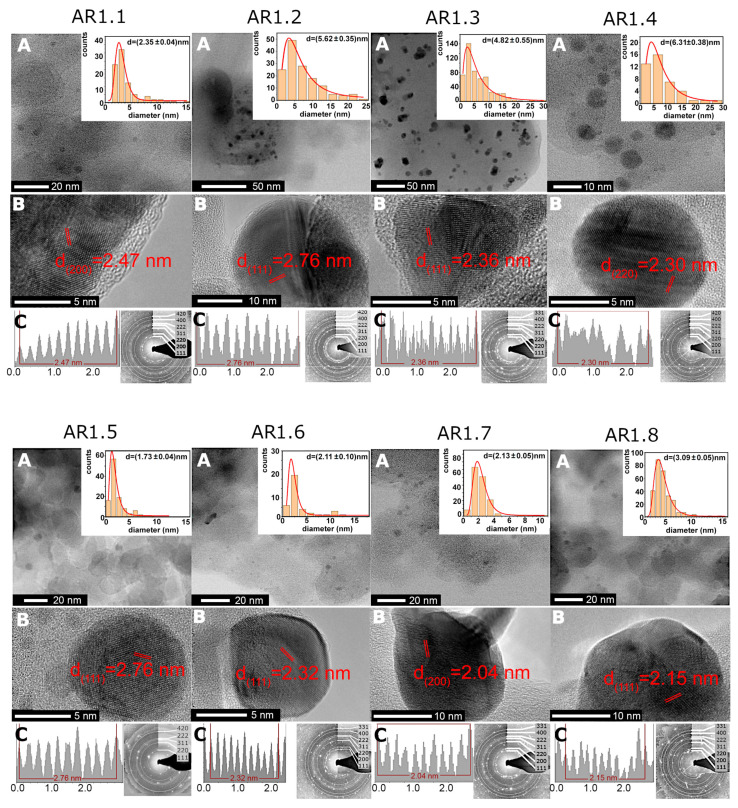
(**A**,**B**) HRTEM microimages for Ag-SiO_2_ nanocomposites (AR1.1–AR1.8) with the inter-planar d-spacing plot along the silver core and the particle size distribution of Ag nanoparticles. The data on histograms were fitted with a logN function. (**C**) Inter-planar spacing plot along the silver nanoparticle core and SAED pattern.

**Figure 4 ijms-24-06632-f004:**
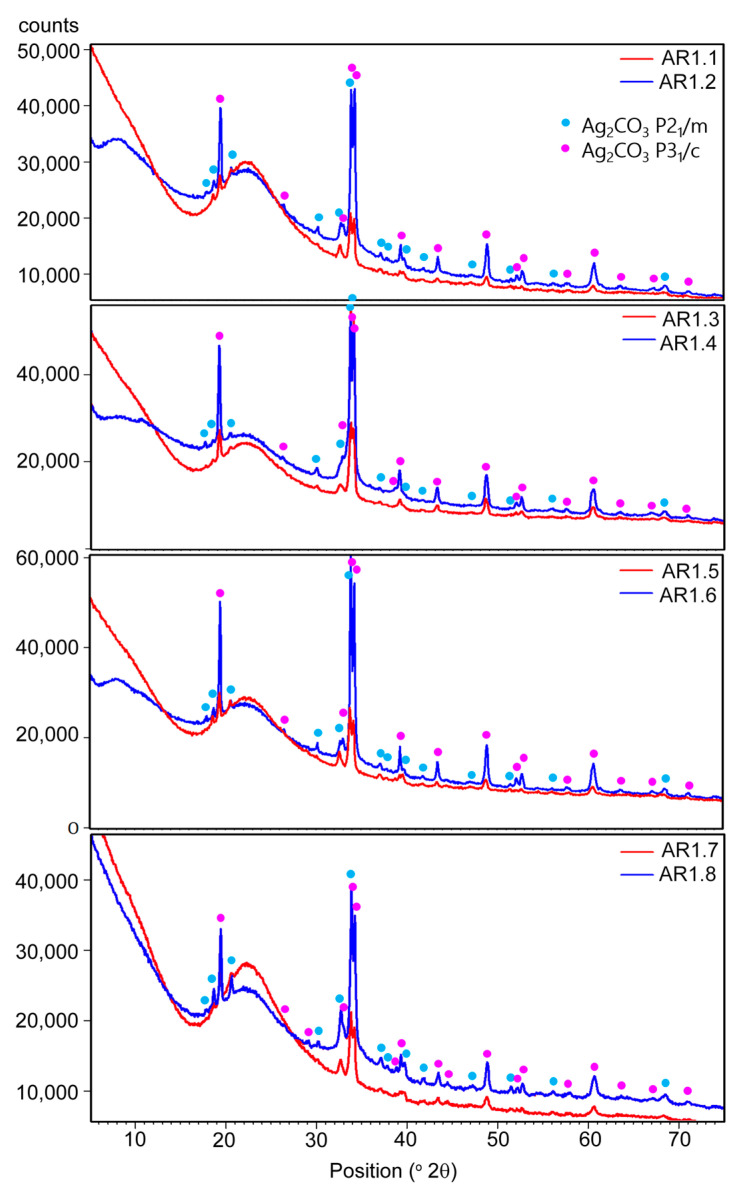
XRD patterns of the Ag-SiO_2_ nanocomposites.

**Figure 5 ijms-24-06632-f005:**
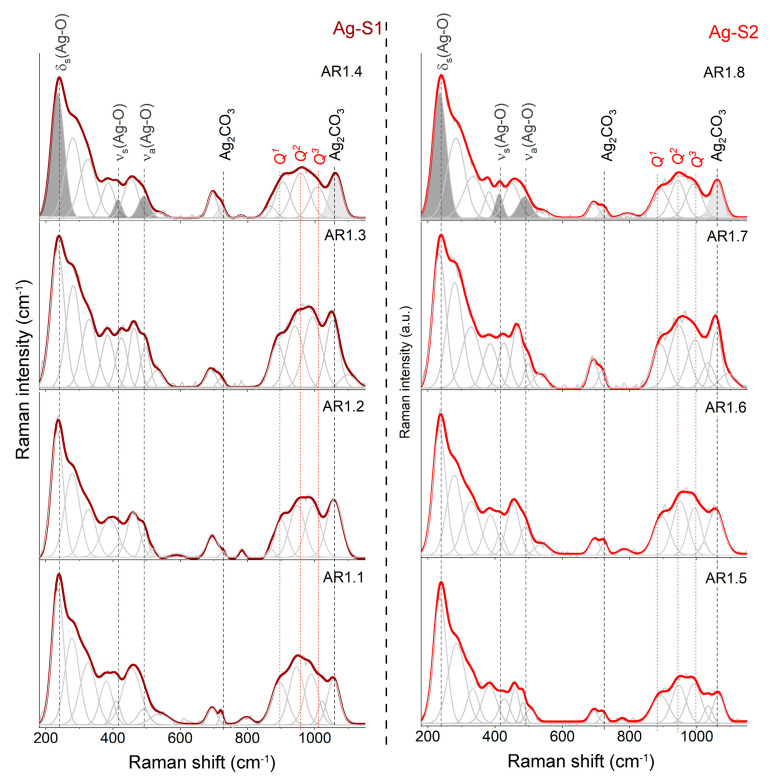
Raman spectra of silver–silica (Ag-SiO_2_) nanocomposites summarized in the 180–1150 cm^−1^ region. Spectra were fitted using the Voigt function at a minimum number of components. *Q*^1^, *Q*^2^, and *Q*^3^ represent individual structural units of amorphous silica. Grey-colored bands are referred to as silver (I and III) oxide (dark) and silver carbonate (light). Ag-S1 and Ag-S2 refer to silver-silica nanocomposites that were prepared on S1 silica (d = 15–20 nm, [53]) and S2 silica (d = 20–30 nm, [54]).

**Figure 6 ijms-24-06632-f006:**
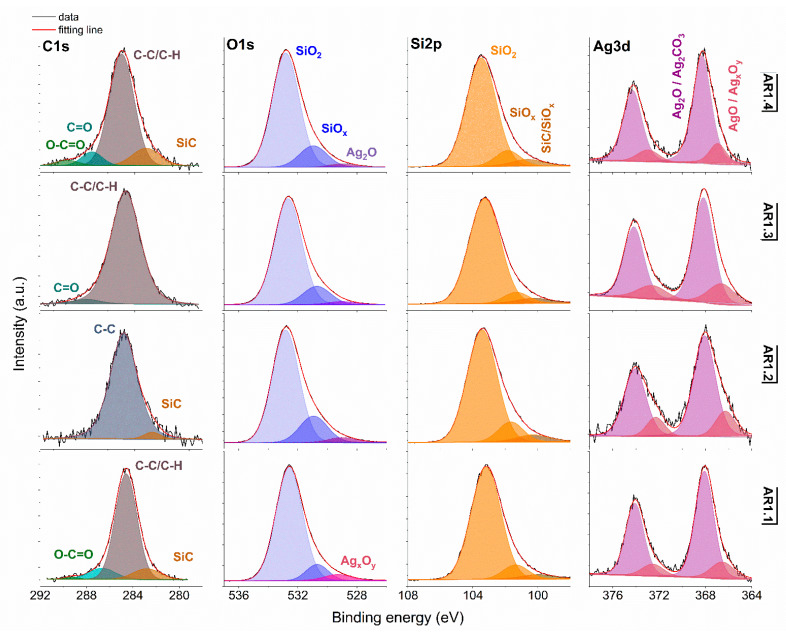
XPS spectra for Ag-S1 with Si 2p, O 1s, Si 2p, and Ag 3d core levels. AR1.1-AR1.4 refer to Ag-S1 silver-silica nanocomposites (S1: d = 15–20 nm, [53]).

**Figure 7 ijms-24-06632-f007:**
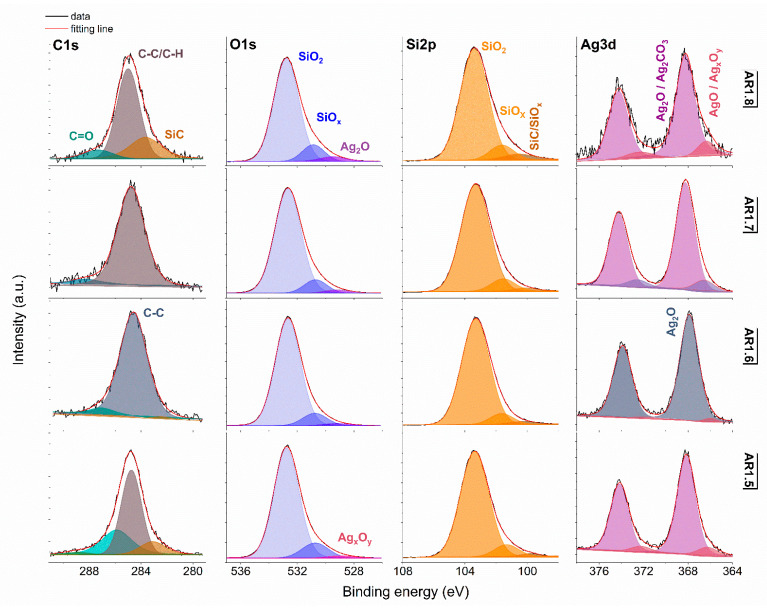
XPS spectra for Ag-S2 with Si 2p, O 1s, Si 2p, and Ag 3d core levels. AR1.5-AR1.8 refer to Ag-S2 silver-silica nanocomposites (S2: d = 20–30 nm, [54]).

**Figure 8 ijms-24-06632-f008:**
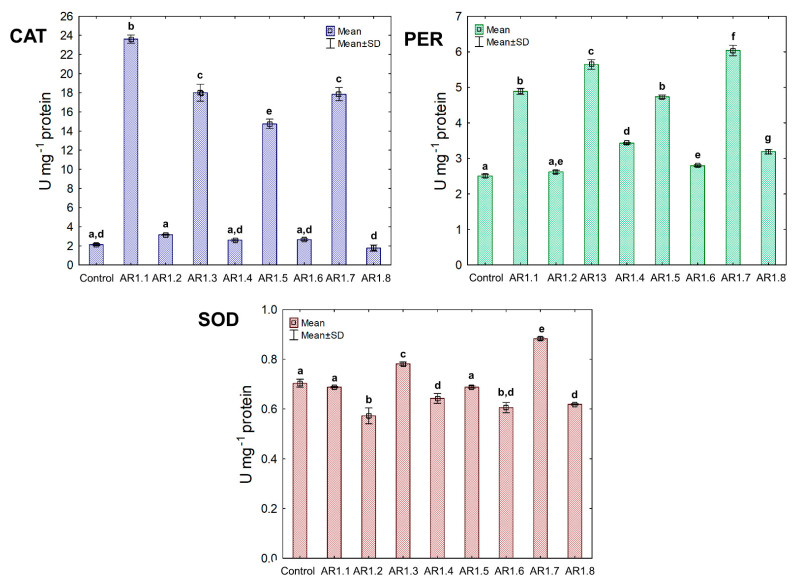
The activities of CAT, PER, and SOD in *E. coli* exposed to NCs at IC_50_ (mean ± SD; n = 3). The same letter(s) indicate no significant statistical differences at *p* < 0.05 among means between NP-treated cells.

**Figure 9 ijms-24-06632-f009:**
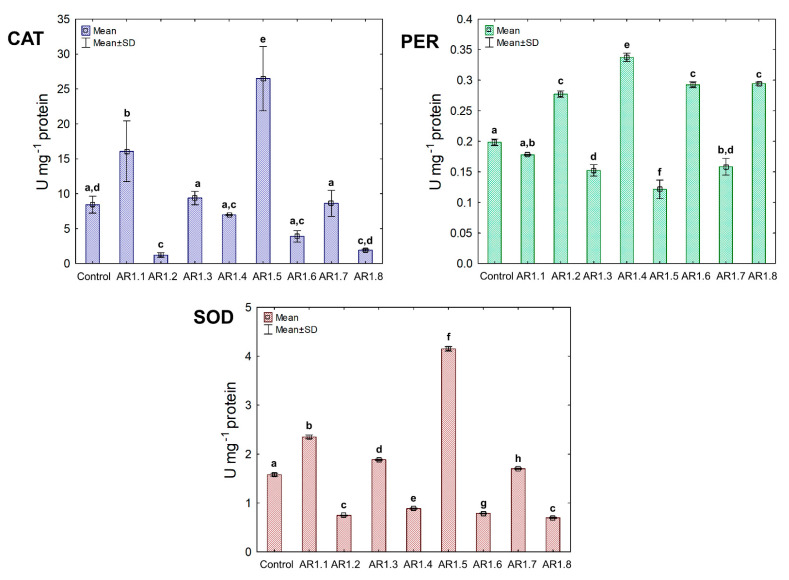
The activities of CAT, PER, and SOD in *B. cereus* exposed to NCs at IC_50_ (mean ± SD; n = 3). The same letter(s) indicate no significant statistical differences at *p* < 0.05 among means between NC-treated cells.

**Figure 10 ijms-24-06632-f010:**
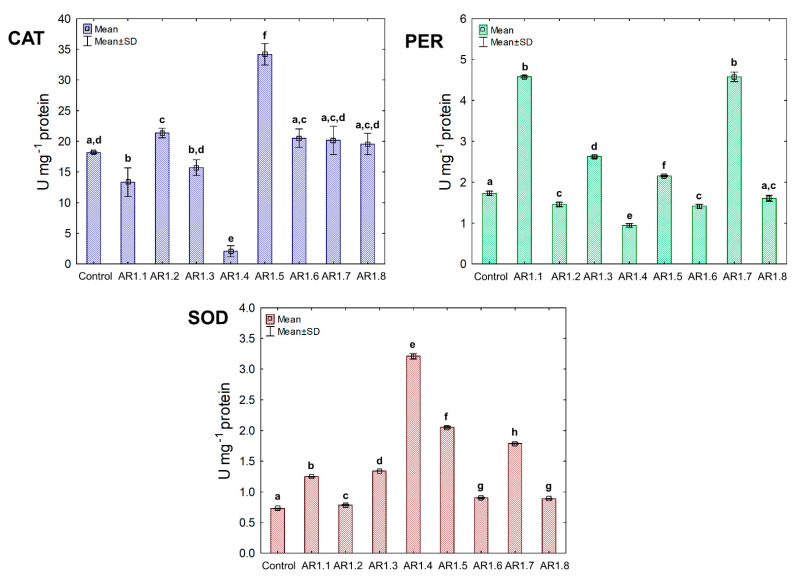
The activities of CAT, PER, and SOD in *S. epidermidis* exposed to NCs at IC_50_ (mean ± SD; n = 3). The same letter(s) indicate no significant statistical differences at *p* < 0.05 among means between NC-treated cells.

**Figure 11 ijms-24-06632-f011:**
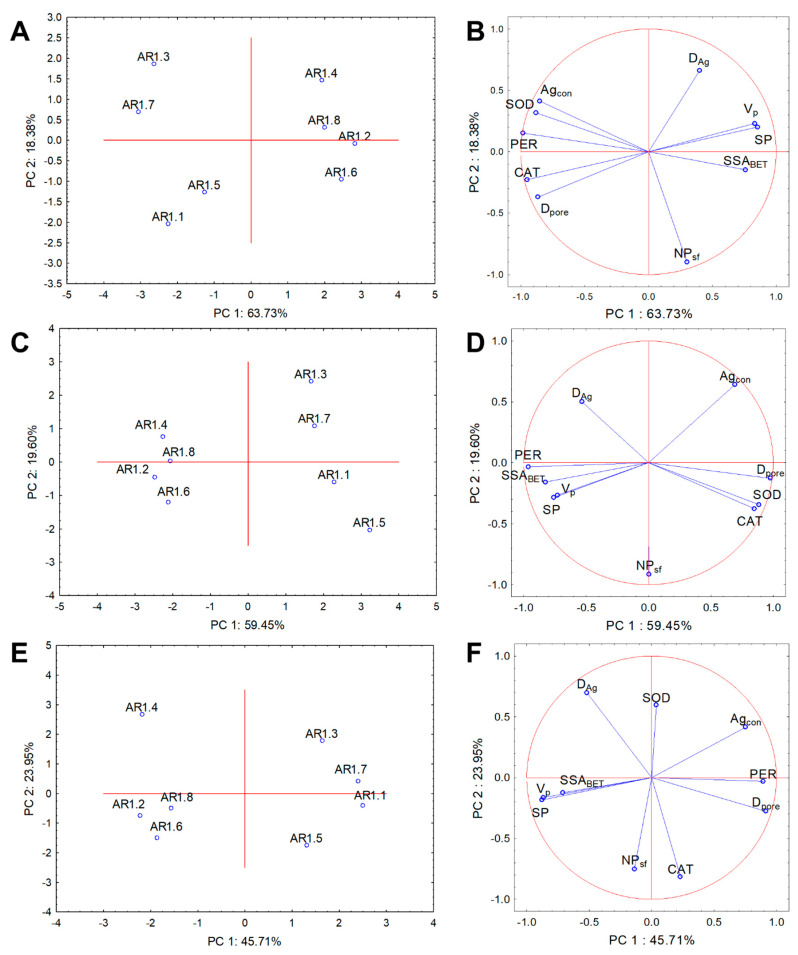
Projection of the individual plots and PCA analysis of CAT, PER, and SOD activity with silver particle diameter (D_Ag_), number of particles with a spherical shape (NP_sf_), pore volume (V_P_), specific surface area (SSA_BET_), micropore surface (S_P_), pore diameter (D_pore_), and silver content (Ag_con_) along PC1 and PC2 for the NP-treated IC_50_ bacterial strains: *E. coli* (**A**,**B**), *B. cereus* (**C**,**D**), and *S. epidermidis* (**E**,**F**).

**Figure 12 ijms-24-06632-f012:**
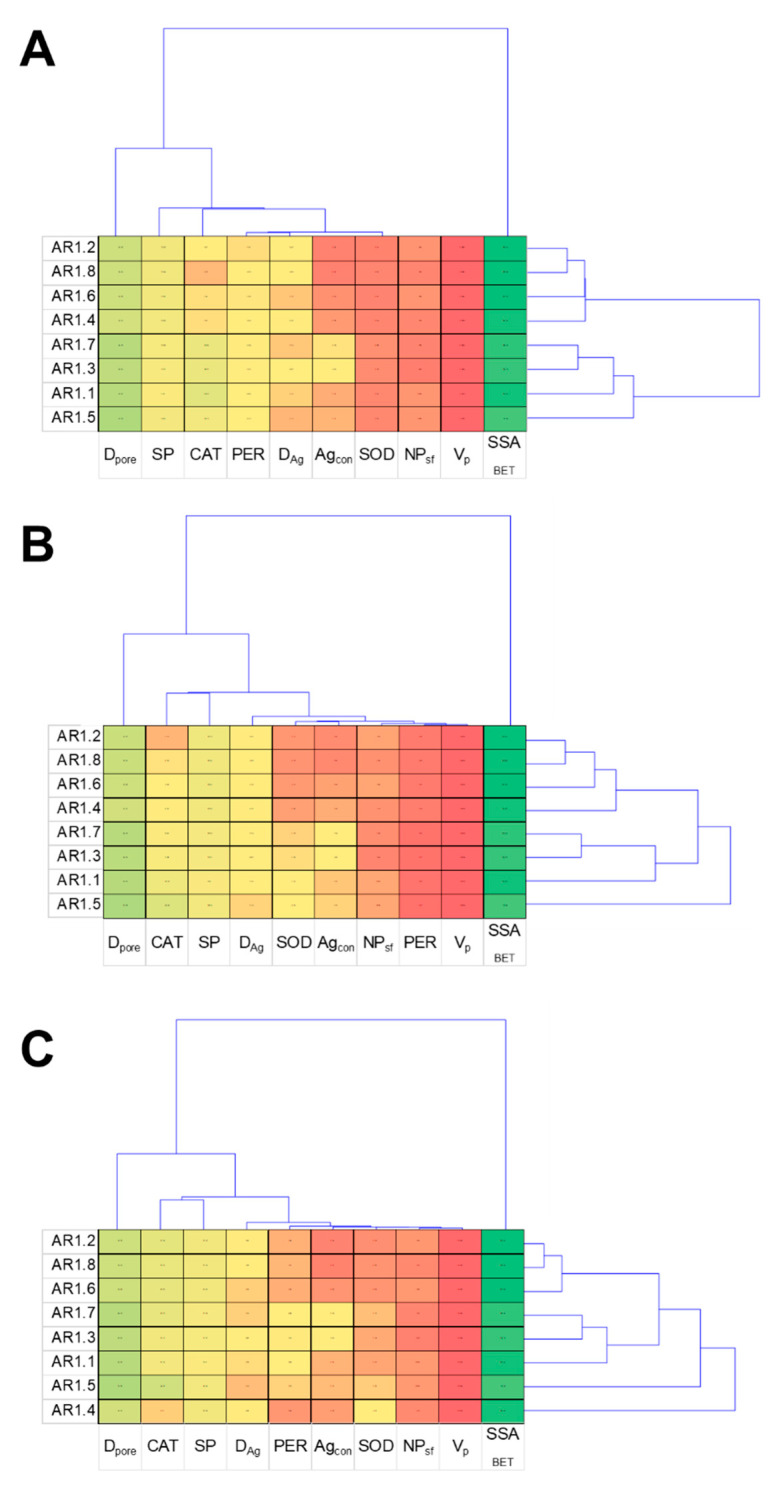
Projection of cluster analysis dendrograms for bacterial strains *E. coli* (**A**), *B. cereus* (**B**), and *S. epidermidis* (**C**) exposed to NCs at IC_50_.

**Table 1 ijms-24-06632-t001:** The samples were prepared at variable synthesis conditions.

	P = 150 W		P = 700 W	
	t = 60 s	t = 150 s	t = 60 s	t = 150 s
Ag-S1	AR1.1	AR1.2	AR1.3	AR1.4
Ag-S2	AR1.5	AR1.6	AR1.7	AR1.8

**Table 2 ijms-24-06632-t002:** Physicochemical properties of Ag-SiO_2_ nanocomposites. d_H_: hydrodynamic diameter, PDI: polydispersity index, and z: Zeta potential. The data are summarized with the standard deviation.

Sample Code	d_H_ (nm)	PDI (-)	z (mV)
AR1.1	146.3 ± 4.8	0.35 ± 0.02	−39.9 ± 0.93
AR1.2	150.9 ± 10.1	0.35 ± 0.03	−37.6 ± 0.86
AR1.3	161.2 ± 3.8	0.33 ± 0.03	−37.5 ± 1.28
AR1.4	154.0 ± 7.5	0.39 ± 0.02	−38.6 ± 1.45
AR1.5	151.4 ± 3.2	0.34 ± 0.03	−38.5 ± 1.99
AR1.6	159.2 ± 10.6	0.37 ± 0.06	−37.2 ± 0.28
AR1.7	140.1 ± 6.1	0.36 ± 0.03	−37.0 ± 1.79
AR1.8	160.8 ± 7.5	0.36 ± 0.05	−38.8 ± 0.65

**Table 3 ijms-24-06632-t003:** Nitrogen sorption parameters for Ag-SiO_2_ prepared at variable microwave power (P = 150 and 700 W) and time (t = 60 and 150 s). SSA_BET_: specific surface area with standard deviation, V_p_: micropore volume, S_p_: micropore surface, D_pore_: pore diameter, TVP: total pore volume, and *: *t*-plot micropore volume. **: *t*-plot micropore area.

Sample Code	SSA_BET_ (m^2^/g)	V_p_ (ml/g) *	S_P_ (m^2^/g) **	D_pore_ (nm)	TPV (cm^3^/g)	Porosity (%)
AR1.1	106.45 ± 0.36	0.00326	9.60	47.02	1.2514	75.02
AR1.2	109.66 ± 0.56	0.00622	15.69	35.78	0.9808	70.19
AR1.3	100.70 ± 0.43	0.00459	11.97	43.56	1.1713	73.76
AR1.4	106.41 ± 0.55	0.00594	15.03	32.37	0.8612	67.39
AR1.5	96.04 ± 0.51	0.00561	14.16	51.17	1.2288	74.68
AR1.6	112.90 ± 0.56	0.00613	15.61	34.99	0.9878	70.33
AR1.7	100.33 ± 0.57	0.00465	12.50	46.69	1.0967	72.47
AR1.8	108.04 ± 0.57	0.00607	15.32	35.76	0.9659	69.86

**Table 4 ijms-24-06632-t004:** Average atomic element concentration for silver–silica nanocomposites estimated using SEM-EDS and TEM-EDS with statistical analysis (mean ± SD, n = 3: SEM, n = 5: TEM).

Sample Code	SEM-EDS			TEM-EDS		
	O (at.%)	Si (at.%)	Ag (at.%)	O (at.%)	Si (at.%)	Ag (at.%)
AR1.1	75.0 (4.6)	21.8 (5.0)	3.2 (1.2)	77.2 (13.1)	12.9 (2.9)	9.9 (1.5)
AR1.2	77.5 (3.2)	20.2 (3.0)	2.4 (1.5)	85.1 (4.6)	14.1 (4.5)	0.8 (0.5)
AR1.3	73.9 (4.5)	23.0 (5.2)	3.1 (0.9)	78.0 (4.9)	19.8 (3.7)	2.2 (1.3)
AR1.4	76.2 (1.0)	20.0 (1.4)	3.8 (0.6)	78.4 (1.0)	21.1 (0.8)	0.5 (0.2)
AR1.5	78.7 (0.7)	18.9 (1.2)	2.4 (0.6)	75.9 (8.7)	15.2 (1.6)	8.9 (3.9)
AR1.6	76.7 (1.8)	20.9 (1.4)	2.5 (0.4)	83.3 (1.9)	16.0 (1.8)	0.6 (0.2)
AR1.7	74.3 (1.5)	24.7 (1.1)	0.9 (0.6)	82.8 (2.4)	16.8 (2.0)	0.4 (0.2)
AR1.8	74.2 (1.2)	23.0 (1.8)	2.8 (1.0)	83.9 (2.8)	16.0 (2.8)	0.1 (0.1)

**Table 5 ijms-24-06632-t005:** The average diameter of silver nanoparticles obtained from lognormal function approximation for Ag-SiO_2_ prepared at variable microwave power (P = 150 and 700 W) and time (t = 60 and 150 s). Silver nanoparticle diameter summarized with statistical error and the percentage content of spherical Ag NPs.

Sample Code	Ag NP Diameter (nm)	% Spherical NPs
AR1.1	2.35 ± 0.04	100%
AR1.2	5.61 ± 0.35	96%
AR1.3	4.82 ± 0.55	46%
AR1.4	6.31 ± 0.38	67%
AR1.5	1.73 ± 0.04	100%
AR1.6	2.10 ± 0.10	98%
AR1.7	2.13 ± 0.05	59%
AR1.8	3.09 ± 0.05	65%

**Table 6 ijms-24-06632-t006:** Lattice parameters of the silver carbonates were determined from the Rietveld refinement.

Phase (Space Group)	Ag_2_CO_3_ (P2_1_)				Ag_2_CO_3_ (P3_1_c)	
Lattice Parameters	a_0_ (Å)	b_0_ (Å)	c_0_ (Å)	β (°)	a_0_ (Å)	c_0_ (Å)
AR1.1	3.255(7)	9.547(7)	4.855(1)	92.20(2)	9.189(8)	6.405(4)
AR1.2	3.258(4)	9.561(5)	4.8656(3)	92.20(7)	9.201(2)	6.411(7)
AR1.3	3.251(7)	9.609(1)	4.876(9)	91.98(2)	9.198(8)	6.405(1)
AR1.4	3.263(6)	9.557(8)	4.866(1)	92.22(2)	9.204(1)	6.421(4)
AR1.5	3.255(9)	9.545(9)	4.863(6)	92.25(1)	9.194(7)	6.404(9)
AR1.6	3.258(9)	9.559(1)	4.861(1)	92.13(6)	9.200(1)	6.413(3)
AR1.7	3.255(9)	9.558(4)	4.864(1)	92.15(2)	9.198(4)	6.406(8)
AR1.8	3.260(1)	9.560(9)	4.865(9)	92.13(1)	9.204(5)	6.414(2)

**Table 7 ijms-24-06632-t007:** Surface atomic concentration values obtained for Ag-SiO_2_ derived from XPS.

	(at.%)			
Sample Code	C 1s	O 1s	Si 2p	Ag 3d
AR 1.1	5.31	63.99	30.42	0.28
AR 1.2	3.55	64.28	32.07	0.10
AR 1.3	2.49	64.15	32.67	0.69
AR 1.4	2.41	65.35	32.03	0.20
AR 1.5	2.97	64.79	31.92	0.32
AR 1.6	4.60	62.79	32.44	0.17
AR 1.7	3.42	64.58	31.49	0.51
AR 1.8	2.58	64.32	33.00	0.10

**Table 8 ijms-24-06632-t008:** The MBC and IC_50_ values of synthesized NCs against *E. coli*, *B. cereus,* and *S. epidermidis*.

	MBC (mg L^−1^)	IC_50_ (mg L^−1^)
AR1.1	25/**10**/25	5.35/**1.83**/7.21
AR1.2	**5**/10/10	**2.11**/5.43/5.95
AR1.3	25/**5**/25	**1.28**/**1.22**/7.97
AR1.4	7.5/**1.5**/10	5.32/6.11/**2**
AR1.5	25/**10**/25	13.30/**5.54**/10.69
AR1.6	**2.5**/10/7.5	**0.31**/0.67/1.69
AR1.7	25/**10**/**10**	12.66/**3.75**/4.87
AR1.8	**5**/10/7.5	1.18/7.79/**0.41**

## Data Availability

Data are stored in the cloud and stick in the form of a backup.

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
