# Peer review of "Microwave Irradiation vs. Structural, Physicochemical, and Biological Features of Porous Environmentally Active Silver–Silica Nanocomposites"

_ijms, 2023, doi:10.3390/ijms24076632_

Round 1

Reviewer 1 Report

The following manuscript entitled “Microwave irradiation vs structural, physicochemical and biological features of porous environmentally active silver-silica nanocomposites” authored by Strach et al. is a have well designed and executed experiments.  They modified the surface of spherical silica using microwave fields to form Ag-SiO2 systems with stable Ag2O/Ag2CO3 heterojunctions. The manuscript is very well constructed, the schematic representation looks nice and the data are well documented. In my opinion, this manuscript can be published in its present form.

Author Response

We want to thank you for your opinion. We are proud that the paper is valuable in the Reviewer's opinion. However, according to the second review, we improved the quality of the text and figures. We modified the Abstract and Introduction to be more transparent and slightly modified the physicochemical part of the text. We also added new information in the microbiological section. We hope the modified version will be the same or even more valuable.

Reviewer 2 Report

This study investigated the microwave irradiation of silver-silica nanocomposites, however, there is lake significance in chemistry and methodology contribution to the field. Thus comprehensive reversion is needed.

1. The most serious problem of this article is the introduction section, too lengthy and misleading to the readers to get the main point of this paper.

2. Detailed information on the materials in the experimental section should be given starting with Materials.

3. More scientific evidence for the toxicity against bacteria experiment is needed.

4. quality of the figures needs improvement.

5. the porosity and specific surface areas for the porous silica and Ag composite nanomaterials are needed. BET cures should be provided.

6. in the abstract, please delete “We constantly seek new solutions to capture heavy metals or degrade toxic components.”

7. The overall language of the manuscript needs to be more scientific.

Author Response

We thank you for the valuable suggestions necessary to improve the paper. According to the Reviewer, we have modified the text, mainly the "Introduction" and "Results and Discussion," to make the text more understandable. The Introduction was rewritten to be more transparent. Similarly, the quality of the Figures was significantly improved, and many images were prepared from the beginning. To sum up, a point-by-point response to the Reviewer's comments has been provided below. Please find hereafter our answers (in red) to the Reviewer's comments (in black). Many modifications in the main text, which had to be essential to improve the paper, were red-markered. English has been verified and proofread by the native speaker.

Round 2

Reviewer 2 Report

The authors have revised the manuscript and it can now be accepted in its present form.